# Anticoagulation for Splanchnic Vein Thrombosis in Myeloproliferative Neoplasms: The Drug and the Duration

**Wafik G. Sedhom [1] and Brady Lee Stein [2],***

1    Department of Medicine, Northwestern University Feinberg School of Medicine, 676 N. St. Clair Street, Suite 2130, Chicago, IL 60611, USA; wafik.sedhom@northwestern.edu

2    Department of Medicine, Division of Hematology/Oncology, Northwestern University Feinberg School of Medicine, 645 N. Michigan Avenue, Suite 1020, Chicago, IL 60611, USA

*    Correspondence: bstein@nm.org

**Abstract:** Myeloproliferative neoplasms are a common cause of splanchnic vein thrombosis, which causes significant morbidity and mortality. Indefinite anticoagulation is the mainstay of therapy, and vitamin K antagonists (VKAs) are routinely used since hematologists have the most experience with this drug class. The role of direct oral anticoagulants (DOACs) is promising, but still undergoing evaluation. Cytoreduction with hydroxyurea or pegylated interferon is often used when cytosis is present, but their roles are yet to be defined when the complete blood count is normal. Janus kinase (JAK) inhibition may have a complementary role in reducing splenomegaly and portal hypertension.

**Keywords:** splanchnic vein thrombosis; myeloproliferative neoplasms; JAK2 V617F mutation; anticoagulation; polycythemia vera





## 1. Introduction

Philadelphia-negative myeloproliferative neoplasms (MPNs) encompass polycythemia vera (PV), essential thrombocythemia (ET), and primary myelofibrosis (MF), which is further divided into pre-fibrotic (pre-MF) and overt MF. Pre-MF is identified as a distinct disease that can mimic features of ET, but is characterized by progression to overt MF and by fibrosis present in the bone marrow [1]. These diseases are characterized by a driver mutation (JAK2 V617F, CALR, or MPL) that leads to excessive proliferation of the myeloid lineage, a potential to transform into acute myeloid leukemia, and an increased risk of arterial and venous thrombosis [2]. A meta-analysis including over 13,000 patients examining thrombosis prevalence around the time of MPN diagnosis revealed arterial thrombosis in 16.2% and venous thrombosis in 6.2% of patients. PV was most commonly complicated by thrombosis (28.6%), as compared to ET (20.7%) and MF (9.5%) [3]. Venous thrombosis in MPNs can include typical sites (e.g., pulmonary embolism, deep venous thrombosis); however, there is a substantial body of evidence pointing to MPNs as a leading cause of splanchnic vein thrombosis (SVT). SVT includes hepatic vein thrombosis (Budd-Chiari syndrome or BCS), mesenteric vein thrombosis (MVT), splenic vein thrombosis, and portal vein thrombosis (PVT). The estimated rates of SVT range from 12% to 39% for PV, 11% to 25% for ET [4], and 7.2% for MF [5]. The aim of this review is to discuss the epidemiology, prognosis, and pathogenesis of SVTs derived from MPNs, as well as to explore established and emerging treatment options.

## 2. Overview of MPN-SVT

### 2.1. Epidemiology

SVTs are most commonly identified in younger patients (age < 45), especially women with PV and low allele-burden JAK2 V617F mutations. Tremblay et al. recently analyzed MPN-associated SVT over 20 years at a single center and identified the mean age of SVT as 45, with 72% of the cohort being female [6]. Another retrospective analysis comparing PV

in patients < 45 years old and >65 years old showed that SVT occurred more frequently in younger patients (13% vs. 2%, *p* = 0.0056), with >88% of those SVT events occurring in women [7].

While obvious myeloproliferation may be present at MPN-SVT diagnosis, it is not uncommon for patients to have an entirely normal complete blood count. A meta-analysis analyzing MPN-associated BCS and PVT identified cases without typical morphological features (i.e., normal CBC) at 17.1% and 15.4%, respectively [8].

PV is the most common MPN to present with SVT, followed by ET, and then MF. One meta-analysis placed PV at 52.9% in BCS and 27.5% in PVT, whereas ET had rates of 24.6% and 26.2%, and MF had rates of 6.7% and 12.8%, respectively [8]. Other thrombophilic disorders have been reported as independent risk factors for the formation of SVT. De Stefano evaluated patients with MPN-SVT and found that inherited thrombophilia was present in one-third of patients. Factor V Leiden was the most common disorder in BCS, and prothrombin G20210A was most common in extrahepatic PVT [9].

Central to MPN disease pathogenesis is the acquisition of driver mutations that result in JAK-STAT dysregulation, the most prevalent mutation being JAK2 V617F. This mutation is also most commonly identified in MPN-SVT, in approximately 80% of cases [10]. Calreticulin (CALR) mutations which affect calcium signaling and protein folding in the endoplasmic reticulum [11] are observed less frequently than JAK2 V617F in SVT. One meta-analysis demonstrated that for all SVT, the pooled prevalence was 1.21%, but in SVTs associated with MPNs, the prevalence increased to 3.71%. Interestingly, the prevalence increased to 15.6% for those MPN-associated SVTs that did not present with a JAK2 V617F mutation [12]. The JAK2 V617F allele burden has also been shown to be different in those with MPN-SVT. In How's study, where allele burden was tested on 10 patients, it was significantly smaller than those MPNs without SVT (median: 5% vs. 36.3%, *p* = 0.019), with no patient reaching 10% [13]. Therefore, JAK2 testing is warranted in those who present with SVT without known MPN and in those with signs of portal hypertension. The testing of CALR mutations, although less prevalent overall, may be beneficial in the workup of JAK2 V617F-negative MPNs. Other mutations have been identified in MPNs, including MPL. This gene codes for the thrombopoietin receptor and can lead to the dysregulation of platelet production. It has been reported to be present in 4% of ET and 5% of pre-MF patients [14], and meta-analysis data of 305 patients with BCS and PVT reported only three cases of MPL W515K mutation (<1%). The prevalence of the mutation in patients with SVT is reported at 0.7%, compared to 33% with JAK2 V617F. Other mutations, such as JAK2 exon 12 mutations, are even less prevalent. Of the 268 patients in the Smalberg meta-analysis, none had aberrations in JAK2 exon 12 [8].

*2.2. Pathogenesis*

2.2.1. MPN-Thrombosis at Large

The multifactorial contributions to MPN thrombosis have been previously summarized, and include advanced age, gender differences, cardiovascular risk factors, inflammatory stress, erythrocytosis, and possibly leukocytosis. In addition, endothelial dysfunction and an abnormal interaction with leukocytes may contribute. The elaboration of the neutrophil elastase traps (NETS) may also contribute [15]. Due to its prevalence, the JAK2 mutation is central to thrombosis in MPNs. It is known, for example, that JAK2 V617F-positive megakaryocytes (in mouse models) demonstrated increased platelet signaling in response to thrombopoietin and have increased chemotaxis (leading to platelet aggregation) in MPN patients [16]. In an analysis of the molecular markers associated with thrombosis, Barbui et al. demonstrated that patients with JAK2 V617F mutations have increased levels of platelet activation (P-selectin, tissue factor, thrombin), endothelial cell activation (thrombomodulin, von Willebrand factor), and leukocyte activation compared to JAK2 V617F-negative patients. These factors contribute to the hypercoagulable environment that predisposes patients to thrombosis [5]. Several other theories have been proposed to identify the pathogenesis of thrombosis in MPN. One posits that an abnormal clonal popu-

lation hyperactivates the coagulation cascade. Another theorizes that the pro-inflammatory nature of the MPN may be responsible for activating the cytokines that activate coagulation factors and cause endothelial injury, one of the central promotors of thrombosis according to Virchow's triad [3]. There are also hypoxia-inducible factor (HIF) genes that have shown upregulation in PV and ET patients. Gangaraju et al. studied several genes in the granulocytes and platelets of PV and ET patients, and found a correlation between the upregulation of these pro-thrombotic genes and the JAK2 V617F allele burden. This upregulation was more prevalent in those with a history of thrombosis [17].

2.2.2. MPN-SVT Pathogenesis

In MPN-SVT, the data indicate a number of factors that could lead to a hypercoagulable environment. An association exists between the JAK2 46/1 haplotype and the development of SVT. Li et al., compiled 26 observational studies and concluded that the JAK2 46/1 haplotype was present in JAK2 V617F-positive and negative MPNs, but the association with SVTs was only present in those with JAK2 V617F positivity [18]. Nitric oxide dysregulation is also believed to lead to SVT in these populations. The dynamics of blood flow were considered in another review, in which splanchnic veins had low flow states which led to localized hyperviscosity. This increases nitric oxide scavenging by the erythrocyte hemoglobin in the red cell mass coming in contact with sinusoidal endothelial cells, which leads to increased thrombogenesis [19,20]. Aberrant endothelial cells themselves may harbor JAK2, as has been demonstrated in liver cells in two BCS patients [21], and also in spleen cells [22] (Figure 1).

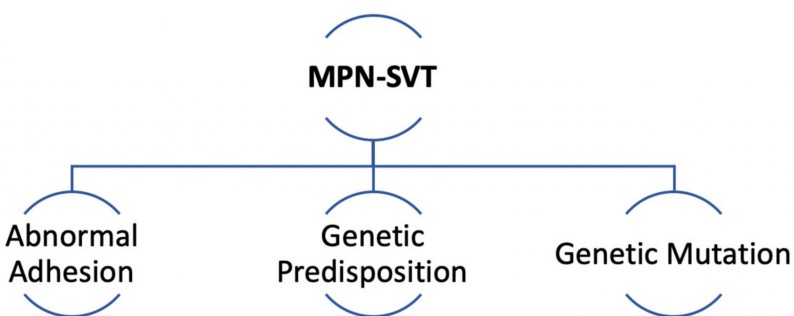

**Figure 1.** The Pathogenesis of Myeloproliferative Neoplasm-associated Splanchnic Vein Thrombosis (MPN-SVT). MPNs lead to the formation of thrombi through a number of mechanisms. Endothelial cells have been known to harbor JAK2, leading to abnormal adhesion. Patients with a JAK2 46/1 haplotype are more susceptible to MPN-SVT [18]. The genetic mutations that promote thrombosis in MPN include JAK2 V617F most frequently, but can also include CALR and MPL. Increased red cell mass in MPNs, particularly PV, can increase nitric oxide scavenging by erythrocyte hemoglobin in low-flow areas such as the splanchnic vasculature [19].

*2.3. Prognosis*

Traditionally, the natural history of the underlying MPN was thought to be the primary driver of prognosis for patients with MPN-SVT. However, emerging data provides more nuance. In an analysis of 3,705 patients with PV or ET, when SVT was the presenting symptom of their disease (118 patients), it was determined that those with SVT lived five years less on average than those without SVT on diagnosis (HR 2.47, $p = 0.001$) [23], after adjusting for age and sex. The primary drivers of the lower life expectancy were liver failure, major bleeding (IRR 3.6, $p < 0.001$), or non-skin related second cancer (IRR 2.37, $p = 0.002$) [23] rather than MPN itself or its transformation to MF or AML. This finding seems to contradict earlier studies, such as a Mayo cohort of 84 patients which concluded that the post-SVT prognosis was determined by the underlying MPN and not affected by the SVT [24], but the authors note that the prior studies had not been age-adjusted. Another study suggested that the mutational profile influenced prognosis—here, investigators noted that patients with high JAK2 V617F allele burdens (50% or greater) fared worse

with regards to progression to myelofibrosis, transformation to acute leukemia, or death (odds ratio of 14.7). The study also identified that patients with non-JAK2 mutations of the spliceosome or chromatin (e.g., TET2, DNMT3A, and ASXL1), or the tumor suppressor gene TP53 also fared worse [25].

### 3. Treatment Choices

The treatment of MPN-associated SVT is complex. Treatment decisions include anti-coagulation (type of agent and duration), cytoreduction (with or without cytosis), management of portal hypertension through transjugular intrahepatic portosystemic shunts (TIPS), and cirrhosis with liver transplant. A multi-disciplinary approach is often needed to manage these patients, including specialists from hematology, gastroenterology, surgery, and interventional radiology.

### 3.1. Anticoagulation

The initial management of MPN-associated SVT should be the same as with any thrombosis: in the absence of any absolute contraindication, prompt anticoagulation with unfractionated heparin (UFH) or low molecular weight heparin (LMWH) can be initiated. Since a common result of SVT is portal hypertension leading to the development of esophageal varices, these must be controlled either with beta-blocker therapy or by endoscopy. After this has been achieved, anticoagulation can be implemented. The mainstay of therapy for these patients has involved bridging to an oral anticoagulant and Vitamin K antagonists (VKAs) such as warfarin have been traditionally used. In a large retrospective cohort of 518 patients with MPN-SVT, anticoagulation was the most effective strategy to reduce recurrence rates: VKA reduced the rate of recurrence by over half (OR 0.48) whereas cytoreduction had a potential for failure (OR 0.96). The bleeding risk of patients was based on whether or not they presented with esophageal varices, which was present in 67% of those studied. Esophageal varices were an independent risk factor for bleeding, with an OR of 17.4. Excluding the variceal bleeding, there was no statistically significant rates of bleeding between the control and the SVT cohorts [2]. Another study looking at the recurrence of thrombosis after the initial SVT event in 181 patients showed that the recurrence per 100 patient-years with or without VKA was 3.7 (95% CI: 2.3–5.5) versus 7.2 (95% CI: 3.1–14.3), respectively ($p = 0.09$) [26].

One recent study of 84 Mayo patients has challenged the notion that VKAs are the ideal treatment strategy. In the study, patients with initial thrombosis were monitored for recurrence while on either anticoagulation, anticoagulation and cytoreduction, anticoagulation and ASA, anticoagulation with ASA and cytoreduction, or no anticoagulation at all. Data indicated the recurrence rate was not affected by the initial treatment strategy [24], which highlights the need to explore other options, such as DOACs or JAK2-inhibitors.

Low molecular weight heparin (LMWH) has been deemed superior to VKA in patients with malignancy-associated venous thrombosis [27]. It is tempting to place patients on LMWH given this justification, but it is important to understand that hematologic malignancy comprised only about 10% of the malignancies studied in these trials [28]. There is little data to suggest that LMWH has superior efficacy over VKAs in MPN-SVT, and given the generally established practice of indefinite anticoagulation for these patients, LMWH injections are less convenient than oral options. LMWH is the treatment choice, however, for pregnant women with active or prior MPN-SVT due to the teratogenicity of VKAs [28]. This is crucial given the high prevalence of SVTs in women of childbearing age.

### 3.2. Direct Oral Anticoagulants (DOACs)

Although VKAs have been the primary treatment for MPN-SVT, they have their own challenges. In patients with BCS and liver failure, for example, INR monitoring may be inaccurate to measure efficacy of anticoagulation [28], particularly if the baseline prothrombin time is prolonged. Direct oral anticoagulants (DOACs) have advantages over VKAs because they do not require monitoring to ensure therapeutic levels and have fewer

bleeding complications. Barbui et al. analyzed patients with MPNs treated with DOACs and compared them to prior studies that utilized VKAs. Fifty-eight of the 239 patients presented with SVT. The rates of thrombosis on DOACs were similar to those on VKAs (4.5% on DOACs vs. 4.7% on VKAs). The rates of bleeding were also similar between the groups (2.3% on DOACs vs. 2.4% on VKAs). Notably, more bleeding was identified in the pre-MF patients and those on dabigatran [29]. Ianotto et al. evaluated DOAC use in 25 patients with PV or ET determined to be high-risk by European Leukemia Net (ELN) criteria. Thirteen patients were treated for atrial fibrillation and twelve patients were treated for thrombotic events. Only one thrombotic event (stroke) was observed on DOACs, with three provoked bleeding events [30].

Huenerbein et al. evaluated 71 MPN patients with either arterial or venous thrombi treated with DOACs (22 of whom had SVT) and found no statistically significant arterial or venous thromboembolism relapse-free survival, but did find fewer relapses on DOACs ($p = 0.0003$) [31]. De Gottardi et al. evaluated the use of DOACs in 94 patients, with 38% having cirrhosis. Within this cohort, 75% of patients had SVT, but none had MPN. One patient had recurrent PVT, and 5 patients had bleeding complications. DOACs were stopped in 3 cases [32]. Although none of these patients had MPN, this study supports the idea that SVTs may be treated effectively and safely with DOACs, regardless of the etiology. Even though these data suggest that DOACs may be promising in SVT, ultimately a large randomized controlled trial comparing DOACs to LMWH or VKAs is needed, since use of a DOAC in these situations is still considered off-label. The clinician should also keep in mind that patients with severe decompensated liver disease (Child–Pugh Class C) cannot use any DOACs and that rivaroxaban is additionally contraindicated for moderately severe liver disease (Child–Pugh Class B) [33].

## 4. Anticoagulation Treatment Duration

### 4.1. MPN Thrombosis at Large

There is a lack of consensus among hematologists with regards to anticoagulation duration for MPN thrombosis at large. A 2014 survey of 73 hematologists, including 42 who considered MPNs their primary area of study, revealed varied approaches to treatment [34]. This highlights the lack of concrete guidelines to approach anticoagulation duration in MPNs in general. The National Comprehensive Cancer Network (NCCN) guidelines indicate that "there are no data to guide the selection or appropriate duration of anticoagulation with or without antiplatelet therapy in patients with PV or ET." NCCN also states that the duration of anticoagulation is reliant on the severity of the inciting event, risk of recurrence, and how well the disease is controlled [35].

There is evidence that indefinite anticoagulation is warranted due to the high rate of recurrent thrombosis in MPN thrombosis at large. In a study of 150 PV and ET patients by Hernandez-Boluda et al., the primary outcome was the recurrence of arterial or venous thrombosis after an initial event for which patients were placed on VKAs. This study found that the rate of re-thrombosis was 4.5 per 100 patient-years, but that rate went up to 12 per 100 patient-years after stopping VKA therapy ($p < 0.0005$) [36]. The data favored an increased risk of thrombosis for those who had a thrombosis history, but interestingly did not find an increased risk for those >60 years of age. The study did not find an increased bleeding risk with anticoagulation. This finding was supported by a study by Wille et al., in which recurrent thrombosis was evaluated in patients with and without VKA therapy. This study identified 99 MPN-associated thrombotic events. When the data was evaluated for all patients who presented with venous thromboembolism (regardless of location), 13 of the 36 patients in the study who had their prophylactic anticoagulation terminated 6 months after the initial event had a VTE recurrence after a median of 13 months, whereas only 3 of the 35 who continued prophylactic anticoagulation developed recurrence [37].

### 4.2. MPN-SVT

Due to the underlying prothrombotic state of MPN and high risk of thrombosis recurrence with SVT, treatment duration has traditionally been indefinite [38], however this is based on consensus opinion rather than prospective studies. Unfortunately, MPN-SVT patients usually comprise a subset of larger retrospective studies noted above. For example, in Hernandez–Boluda's study of 150 patients, 20 patients had portosplenic/mesenteric vein thrombi and 17 had BCS. In Wille's study, 32.3% of the 99 thrombotic events involved the splanchnic vasculature. The largest MPN-SVT cohort, presented by Sant'Antonio (N = 518) reported a significant reduction in thrombosis recurrence with anticoagulation [2]. It is challenging to extrapolate results, but in general, provided the benefits outweigh the risks, indefinite anticoagulation is typically advised for MPN-SVT patients.

## 5. Cytoreduction

### 5.1. Hematocrit Control

The CYTO-PV trial established that there was a benefit to lower hematocrit goals in JAK2-positive PV patients (hematocrit <45% compared to 45–50%) to reduce the risk of thrombotic complications [39]. Cytoreduction through phlebotomy has been a mainstay of treatment for PV patients that have hematocrits above this threshold. However, it is noted that some patients with MPN-SVT can have entirely normal hematocrits (Smalberg's landmark meta-analysis notes normal CBCs in 17.1% and 15.4% of their BCS and PVT populations, respectively [8]). In some, the increased red cell mass that would be expected can be masked by the presence of portal hypertension, making it appear as if the patient has a normal hematocrit [40]. It is unknown if patients with normal CBC require cytoreduction. No high-quality evidence currently exists that patients with a normal CBC need cytoreduction to reduce the risk of thrombosis.

### 5.2. Hydroxyurea

Based on the current guidelines from European Leukemia Net and NCCN, hydroxyurea (HU) or pegylated interferon alpha are the first-line therapies for cytoreduction in ET and PV [29]. In a trial encompassing over 1500 patients examining hydroxyurea in combination with antiplatelet and VKA for the prevention of arterial and venous thrombi (of which 218 patients had SVT), hydroxyurea did not reduce recurrent SVT, although it did reduce recurrent arterial thrombi [41]. The authors hypothesized that different mechanisms may have been involved in the pathogenesis of thrombi at these different sites, such as the aberrant endothelial cells that may carry the JAK2 V617F mutation. The pathogenesis may also be less reliant on hypercythemia, meaning that cytoreduction would be of little utility. Their ultimate recommendation was only to use HU in the presence of hypercythemia or progressive disease.

### 5.3. Pegylated Interferon

For patients who are intolerant of or resistant to hydroxyurea, pegylated interferon therapy (PEG-IFN) is an option. It may also be used as first line therapy in some instances. PEG-IFN is thought to work by reducing JAK2 V617F-positive stem cells [42], which release prothrombotic cytokines. In a 20-patient study of PV and ET patients with prior SVT, PEG was initiated with the goal of observing the overall hematological response rate (ORR, CR + PR) and recurrence of SVT. The ORR was 70% after 12 months of therapy. None of the patients had an SVT recurrence after 2.2 years, although this data may be confounded since 90% of the study patients were on anticoagulation/antiplatelet therapy. However, there was no reduction in spleen length by ultrasound in the 9 patients who completed the treatment [43].

### 5.4. Ruxolitinib

Ruxolitinib, a JAK1/2 inhibitor, has been approved for the treatment of hydroxyurea-resistant PV, and for MF [44]. Its use was studied by Pieri et al. in a Phase 2 clinical trial for

patients with MPN-SVT. The primary outcomes were the reduction of splenomegaly >35% by MRI/CT or >50% by palpation after 24 weeks of therapy. There was, on average, a 28% reduction in spleen size by MRI/CT and 38.2% by palpation. The data also suggested a stabilization of esophageal varices at 72 weeks due to the overall reduction in portal hypertension. Additionally, only one bleeding episode was reported in the 21 patients [37], indicating the treatment is generally well tolerated in those with SVT and has a role in alleviating burdensome symptoms. Another JAK2 inhibitor, fedratinib, has also recently been FDA-approved for MF, but studies in the SVT population have yet to be performed.

## 6. Conclusions

Myeloproliferative neoplasms come with an increased risk of thrombosis, and splanchnic venous thrombosis is a unique complication. It most commonly affects younger women, and can be present even in the absence of erythrocytosis or thrombocytosis. An essential driver mutation, most commonly JAK2 V617F, promotes a thrombotic environment. For patients with MPNs, the presence of SVTs can predict a worse prognosis compared to those without. Treatment modalities have included VKAs, but emerging data for DOACs suggest they may be safe for use. Treatment duration is based on expert consensus, but given the highly thrombotic nature and risk of recurrence, indefinite anticoagulation is justified as long as the benefits outweigh the risks. Cytoreduction using hydroxyurea or pegylated interferon is suggested for MPNs with hypercythemia or progressive disease to reduce the risk of thrombosis. The JAK2 inhibitor ruxolitinib can be utilized in patients with refractory splenomegaly. Unanswered questions remain, including which cytoreductive agent is superior, if cytoreduction should be used with normal blood counts, and if DOACs are superior to VKA.

**Author Contributions:** Writing—original draft preparation, W.G.S.; writing—review and editing, B.L.S. All authors have read and agreed to the published version of the manuscript.

**Funding:** This research received no external funding.

**Institutional Review Board Statement:** Not applicable.

**Informed Consent Statement:** Not applicable.

**Data Availability Statement:** Not applicable.

**Conflicts of Interest:** B.L.S. has consulted for Constellation Pharmaceuticals and Pharmaessentia.

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
