# Peer review of "Anticoagulation for Splanchnic Vein Thrombosis in Myeloproliferative Neoplasms: The Drug and the Duration"

_hemato, doi:10.3390/hemato2020015_

Round 1
Reviewer 1 Report
Critique
This is timely and well-written review of SVT in MPNs.
When discussing the mechanism of thrombosis in MPNs authors omitted that in PV and ET there is an increased HIF activity and that some of the thrombosis regulating genes are HIF-regulated (tissue factor, protein S, etc.) see Gangaraju Blood Advances last year.
When discussing the association of JAK2 V617F allelic burden and SVTs, Steve Oh paper on JAK2 V617F mutated SVTs with normal CBC should be discussed.
For non-MPN expert few sentences are needed to introduce prefibrotic MF as most of these pts are still diagnosed as ET.
Minor comments:
- SVTs are most commonly identified in younger patients, (age < 45) especially women 40 “ give here relative rate of SVTs among young women and men, although you discuss it later but without directly comparing gender and age.
- Central to the pathogenesis of thrombosis in MPNs in general is the presence of the JAK2 mutation given prevalence.” Please reword
- the data indicates. “ Data is plural, datum is singular.
- Legend Fig.1: This increased red cell mass is accompanied by an expanded plasma volume, which may mask the erythrocytosis and lead to a falsely normal hematocrit.” Delete this sentence, it is not relevant to thrombosis and this figure.
- Child-Hugh 197 Class C” needs to be defined
- to approach AC duration” acronym not defined
- Ruxolitinib, a JAK1/2 inhibitor, has been approved for treatment of HU resistant PV, and for MF [37].” Once you state this, I think it is mandatory that you should at least say that also fedratinib it also FDA approved for MF.
Author Response
Please see the attachment.
- When discussing the mechanism of thrombosis in MPNs authors omitted that in PV and ET there is an increased HIF activity and that some of the thrombosis regulating genes are HIF-regulated (tissue factor, protein S, etc.) see Gangaraju Blood Advances last year.
Response: Added “There are also hypoxia-inducible factor (HIF) genes that have shown upregulation in PV and ET patients. Gangaraju et al studied several genes in the granulocytes and platelets of PV and ET patients, and found correlation between upregulation of these pro-thrombotic genes and JAK2 V617F allele burden. The upregulation was more prevalent in those with a history of thrombosis [15].”
- When discussing the association of JAK2 V617F allelic burden and SVTs, Steve Oh paper on JAK2 V617F mutated SVTs with normal CBC should be discussed.
Response: Added “The JAK2 V617F allele burden has also been shown to be different in those with MPN-SVT as well. In How’s study, where allele burden was tested on 10 patients, it was significantly smaller than those MPNs without SVT (median: 5% vs. 36.3%, P = 0.019), with no patient reaching 10% [13]”
- For non-MPN expert few sentences are needed to introduce prefibrotic MF as most of these patients are still diagnosed as ET.
Response: Added “Pre-MF is identified as a distinct disease that can mimic features of ET, but is characterized by progression to overt MF and by fibrosis present in the bone marrow [1].”
Minor comments:
- SVTs are most commonly identified in younger patients, (age < 45) especially women 40 “give here relative rate of SVTs among young women and men, although you discuss it later but without directly comparing gender and age.
Response: Restructured the paragraph to more closely associate the opening sentence with the supporting data: “SVTs are most commonly identified in younger patients, (age < 45) especially women with PV and low allele burden JAK2 mutations. Tremblay et al recently analyzed MPN-associated SVT over 20 years at a single center and identified the mean age of SVT at 45, with 72% of the cohort being female [6]. Another retrospective analysis comparing PV in patients <45 years old and >65 years old showed that SVT occurred more frequently in younger patients (13% vs. 2%, p = 0.0056), with >88% of those SVT events occurring in women [7].”
- Central to the pathogenesis of thrombosis in MPNs in general is the presence of the JAK2 mutation given prevalence.” Please reword
Response: Changed sentence to read: “Due to its prevalence, the JAK2 mutation is central to thrombosis in MPNs.”
- the data indicates. “Data is plural, datum is singular.
Response: Changed to “the data indicate”
- Legend Fig.1: This increased red cell mass is accompanied by an expanded plasma volume, which may mask the erythrocytosis and lead to a falsely normal hematocrit.” Delete this sentence, it is not relevant to thrombosis and this figure.
Response: Removed this sentence from the figure legend and it’s corresponding bubble in the figure.
- Child-Hugh 197 Class C” needs to be defined.
Response: Corrected Child-Hugh to Child-Pugh and changed the sentence to read: “The clinician should also keep in mind that patients with severe decompensated liver disease (Child-Pugh Class C) cannot use any DOACs and that rivaroxaban is additionally contraindicated for moderately severe liver disease (Child-Pugh Class B)[29].”
- to approach AC duration” acronym not defined
Response: Replaced “AC” with anticoagulation
- Ruxolitinib, a JAK1/2 inhibitor, has been approved for treatment of HU resistant PV, and for MF [37].” Once you state this, I think it is mandatory that you should at least say that also fedratinib it also FDA approved for MF.
Response: Added “Another JAK2 inhibitor, fedratinib, has also recently been FDA-approved for MF, but studies in the SVT population have yet to be done.”

Reviewer 2 Report
Dear Editor in Chief,
Sedhom and Stein proposed us a review on “Anticoagulation for unusual site thrombosis in myelo-proliferative neoplasms: the drug and the duration”. This review is very interesting and serious. Please, find enclosed my comments and questions.
In general:
- the title is clearly announcing “unusual site thrombosis“: for the authors, what are these unusual sites? Only SVT? What about CVT for example? As no other sites are described in this review: the title should be modified to make appear “SVT” or “peri-hepatic thromboses” to clarify the purpose?
- in many sections, the informations are mixed concerning thrombosis and treatments in general and specifically in SVT sites, as many data are lacking in the last specific domain. Could the authors more explicitly separate data to improve the lecture?
Introduction:
- line 22: please write “primary” MF rather than MF (because in “secondary” MF there is no PreMF or over-MF). As you known, PMF is a specific MPN entity not secondary MF.
Overview of MPN-SVT:
Epidemiology:
- As supposed, we will expect to read first about incidence or prevalence of SVT in MPN and reciprocally.
- lines 40-41: maybe this sentence should be removed from the beginning since the profile is only described by using Tremblay article, beginning at line 45.
- lines 41-43: talking about normal blood count is a major issue in MPN with SVT, it will be more appropriate in a specific paragraph.
- line 59: 1 line about CALR role (as JAK2, line 58 and MPL, lines 67-68).
Pathogenesis:
- lines 86-89: does the Ref 2 summarized all the propositions written in these 4 lines?
Treatment choices: “anticoagulant” treatment choices?
Anticoagulation:
- line 171: please insert here the fact that most of the cases are diagnosed in young women with childbearing capacity, as written lines 45-46.
DOACs:
- lines 178 to 189 and 190 to 194: first part is about MPN patients and DOAC, second part is about SVT and DOAC without MPN patients. To be clearer, please separate these two parts in two paragraphs.
Treatment duration: “anticoagulant” treatment duration?
- line 204: “AC” means “anticoagulation”, I supposed?
- line 212: use “recurrent” rather than “repeat” thrombosis.
- lines 212 to 227: please write some words about Ref 2, from Sant’Antonio, specifically published on this topic (with 518 pts) and a major conclusion: “never stop VKA in MPN!”.
Cytoreduction:
- part 1 : HU? because of part 2 PEG-IFN and part 3 Ruxolitinib?
- Please discuss some lines about the high rate of normal blood counts observed in these specific cases/the objectives of ELN response/the CYTO-PV results and the fact that all the patients do not correspond to classic proliferative MPN profile and need the same objectives of cytoreduction (Ht<45% versus Ht <42%, Leucocytes <5 giga/l or platelets <200 giga/l)…
- some informations about phlebotomies, as many patients have PV?
Author Response
Please see the attachment.
In General:
- the title is clearly announcing “unusual site thrombosis” for the authors, what are these unusual sites? Only SVT? What about CVT for example? As no other sites are described in this review: the title should be modified to make appear “SVT” or “peri-hepatic thromboses” to clarify the purpose?
Response: Changed the title to: “Anticoagulation for splanchnic vein thrombosis in myeloproliferative neoplasms: the drug and the duration”
- In many sections, the information is mixed concerning thrombosis and treatments in general and specifically in SVT sites, as many data are lacking in the last specific domain. Could the authors more explicitly separate data to improve the lecture?
Response: Several sections have been separated with headings “MPN thrombosis at large” and “MPN-SVT”
Introduction:
- line 22: please write “primary” MF rather than MF (because in “secondary” MF there is no PreMF or over-MF). As you known, PMF is a specific MPN entity not secondary MF.
Response: Added “primary” to MF
Overview of MPN-SVT:
Epidemiology:
- Lines 40-41: maybe this sentence should be removed from the beginning since the profile is only described by using Tremblay article, beginning at line 45.
Response: Restructured the paragraph to more closely associate the opening sentence with the supporting data: “SVTs are most commonly identified in younger patients, (age < 45) especially women with PV and low allele burden JAK2 mutations. Tremblay et al recently analyzed MPN-associated SVT…”
- Lines 41-43: talking about normal blood count is a major issue in MPN with SVT, it will be more appropriate in a specific paragraph.
Response: Placed this information into a specific paragraph.
- Line 59: 1 line about CALR role (as JAK2, line 58 and MPL, lines 67-68).
Response: Added “Calreticulin (CALR) mutations, which affect calcium signaling and protein folding in the endoplasmic reticulum [10], are observed less frequently than JAK2 V617F in SVT.”
Pathogenesis:
- Lines 86-89: does the Ref 2 summarized all the propositions written in these 4 lines?
Response: Yes, it does.
Treatment choices:
- “anticoagulant” treatment choices?
Response: The opening paragraph discusses general treatment choices, not specifically anticoagulation, so we did not make modifications to this.
Anticoagulation:
- Line 171: please insert here the fact that most of the cases are diagnosed in young women with childbearing capacity, as written lines 45-46.
Response: Added “This is crucial given the high prevalence of SVTs in women of childbearing age.”
DOACs:
- lines 178 to 189 and 190 to 194: first part is about MPN patients and DOAC, second part is about SVT and DOAC without MPN patients. To be clearer, please separate these two parts in two paragraphs.
Response: Separated the two paragraphs.
Treatment duration:
- “anticoagulant” treatment duration?
Response: Added “anticoagulation” to treatment duration
- Line 204: “AC” means “anticoagulation”, I supposed?
Response: Replaced “AC” with anticoagulation
- Line 212: use “recurrent” rather than “repeat” thrombosis.
Response: Replaced “repeat” with “recurrent”
- Lines 212 to 227: please write some words about Ref 2, from Sant’Antonio, specifically published on this topic (with 518 pts) and a major conclusion: “never stop VKA in MPN!”.
Response: Added “The Sant’Antonio cohort of 518 patients ultimately saw a trend toward venous thromboses in MPN-SVT. That, in combination with the previously described reduction in recurrence by over half (OR 0.48) leads to the conclusion that continuation of VKAs indefinitely leads to better clinical outcomes [1].”
Cytoreduction:
- Part 1: HU? because of part 2 PEG-IFN and part 3 Ruxolitinib?
Response: Made hydroxyurea its own paragraph
- Please discuss some lines about the high rate of normal blood counts observed in these specific cases/the objectives of ELN response/the CYTO-PV results and the fact that all the patients do not correspond to classic proliferative MPN profile and need the same objectives of cytoreduction (Ht<45% versusHt <42%, Leucocytes <5 giga/l or platelets <200 giga/l.
Response: Added paragraph “The CYTO-PV trial established that there was benefit to lower hematocrit goals in JAK2-positive PV patients (hematocrit <45% compared to 45-50%) to reduce the risk of thrombotic complications [37]. Cytoreduction through phlebotomy has been a mainstay of treatment for PV patients that have hematocrits above this threshold. However, it is noted that some patients can have entirely normal hematocrits (Smalberg’s landmark meta-analysis notes normal CBCs in 17.1% and 15.4% in their BCS and PVT populations, respectively [8]). It is unknown if patients with normal CBC require cytoreduction. No high-quality evidence currently exists that patients with a normal CBC need cytoreduction to reduce the risk of thrombosis.”
- Some information about phlebotomies, as many patients have PV?
Response: This information has been added in the paragraph which is a response to point number 16 above.

Round 2
Reviewer 2 Report
Authors have upgraded the manuscript as asked.